# From Molecules to Mixtures: Learning Representations of Olfactory Mixture Similarity using Inductive Biases

## Abstract

Olfaction—how molecules are perceived as odors to humans—remains poorly understood. Recently, the principal odor map (POM) was introduced to digitize the olfactory properties of single compounds. However, smells in real life are not pure single molecules, but complex mixtures of molecules, whose representations remain relatively under-explored. In this work, we introduce POMMix, an extension of the POM to represent mixtures. Our representation builds upon the symmetries of the problem space in a hierarchical manner: (1) graph neural networks for building molecular embeddings, (2) attention mechanisms for aggregating molecular representations into mixture representations, and (3) cosine prediction heads to encode olfactory perceptual distance in the mixture embedding space. POMMix achieves state-of-the-art predictive performance across multiple datasets. We also evaluate the generalizability of the representation on multiple splits when applied to unseen molecules and mixture sizes. Our work advances the effort to digitize olfaction, and highlights the synergy of domain expertise and deep learning in crafting expressive representations in low-data regimes.

## 1 Introduction

A central challenge in neuroscience is deciphering the link between the physical properties of a stimulus and its perceptual characteristics. While this relationship is well-defined for senses like vision (wavelength to color) and audition (frequency to pitch), it remains elusive for olfaction, a chemical sense, where the mapping from chemical structure to odor perception is complex and not fully understood (Sell, 2006; Barwich & Lloyd, 2022; Barwich, 2022). A recent advance towards digitizing olfaction came with the introduction of the Principal Odor Map (POM) by Lee et al. (2023), a high-dimensional, data-driven representation of odor perceptual space learned from molecular structures. This model demonstrated human-level performance in predicting odor qualities of single molecules and generalized well to other olfactory tasks. However, naturally occurring olfactory stimuli are not comprised of single molecules, but rather complex mixtures of molecules, whose representations remain relatively unexplored within the existing literature. This work introduces POMMix—a mixture and distance-aware extension of the POM representation.

A searchable, rankable, and optimizable digital representation of olfactory space has potential applications in diverse areas (Spence et al., 2017). Such a representation could be used to develop mosquito repellents (Wei et al., 2024), inform agricultural practices by enabling targeted manipulation of insect behavior (Conchou et al., 2019), improve food quality and reduce waste through enhanced spoilage detection (Jung et al., 2023), and accelerate the design of novel fragrance and flavor compounds, which is particularly valuable given increasing regulatory constraints on existing ingredients (Demyttenaere, 2012; IFRA, 2024).

Deep learning models enable the construction of task-optimized data representations, learning complex relationships directly from data (Bengio et al., 2012). However, in low-data regimes, the success of deep learning hinges on incorporating appropriate inductive biases, effectively injecting domain-specific knowledge to guide the learning process and improve generalization (Tom et al., 2023). Olfactory data is currently in this regime—gathering olfactory data is expensive and labor-intensive as it requires training human panelists, filtering potentially toxic molecules, and navigating ethi-

cal review boards. Furthermore, probing human perception is inherently complex, necessitating rigorous data collection protocols and large sample sizes to mitigate individual biases. Existing work on olfactory mixture modeling remains limited, employing a small pool of compounds with low coverage of chemical space (Ravia et al., 2020; Snitz et al., 2019; Sisson et al., 2023; Snitz et al., 2013). While the perfume industry reportedly utilizes 10,000–20,000 compounds routinely, the largest publicly available dataset GoodScents–Leffingwell (GS-LF) contains only around 5,000 molecules (Sanchez-Lengeling et al., 2019; Lee et al., 2023). Significantly larger repositories of mixture characterizations (blends and perfumes) exist within the industry, but remain inaccessible behind private doors.

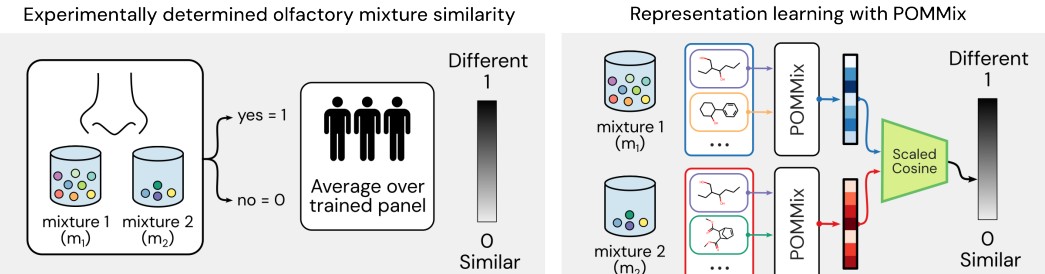

Figure 1: **Task schematic**. Data collection process for olfactory mixture similarities (left), and our approach to predicting olfactory mixture similarities (right).

POMMIX is built by training a neural network to tackle the mixture similarity problem by jointly training a POM with an attention-based mixture model to predict the perceptual similarity of mixtures. This approach also allows us to combine mono-molecular datasets (up to 5,000 data points) and more limited mixture data sources (up to 1,000 data points).

At each stage of our work, we take care to respect the natural symmetries of the problem space—namely, the permutation invariance of molecular descriptions (introduced by the graph representation of a molecule), the permutation invariance of mixture compositions (the model should not care what order the mixture ingredients are presented), and the symmetry of mixture similarities (the model should predict that the similarity of mixture 1 and mixture 2 is the same as the similarity for mixture 2 and mixture 1). The end result is an extension of the POM to mixtures, and a new model building block, dubbed CHEMIX, for encoding mixtures of molecules.

## 1.1 LIST OF CONTRIBUTIONS

- We introduce POMMIX, the first extension of the POM to predict the olfactory similarities between mixtures of molecules.
- We compiled and comprehensively analyzed the limited publicly available olfactory mixture perception data.
- Our model takes into account the inductive biases of the problem and achieves state-of-the-art predictive performance.
- We test our representation in several olfactory settings: the olfactory white-noise hypothesis, generalization to unseen molecules and mixture sizes, and a qualitative study of the interpretability of components within mixtures.
- We make our data and code available for future extensions of our work and for reproducibility: https://anonymous.4open.science/r/anon-iclr2025-DE62.

## 1.2 RELATED WORKS

The modeling of molecular structure-property relationships has a rich history. Within the olfactory domain, previous contributions have utilized hand-picked expert descriptors with classical machine learning algorithms (e.g. tree-based models, support vector machine, and linear models), and/or similarity measures (e.g., cosine, angle) (Snitz et al., 2013; Keller et al., 2017; Kowalewski & Ray,

2020; Vigneau et al., 2018). More recently, deep learning based models have been actively explored to create a more expressive molecular representation of olfactory space (Lee et al., 2023; Tran et al., 2018; Zhang et al., 2024; Sisson, 2022; Maziarka et al., 2020).

Deep learning techniques have been explored in modeling molecular structure-property relationships, including variational autoencoders (Gómez-Bombarelli et al., 2018; Oliveira et al., 2022), large language models (Chithrananda et al., 2020; Ross et al., 2022), and graph neural networks (GNNs) (Yang et al., 2019; Wang et al., 2021). Graph neural networks and graph attention networks (GANs) (Heid et al., 2024; Wu et al., 2023; Buterez et al., 2024) in particular have shown state-of-the-art performance in many molecular property prediction tasks including modeling olfactory space.

Olfactory mixture property prediction is a much more difficult task with fewer effective attempts (Lapid et al., 2008; Khan et al., 2007; Olsson, 1998; Dhurandhar et al., 2023; Ravia et al., 2020). Molecular mixtures have been studied before for battery electrolytes by Zhang et al. (2023). The work, however, uses a large dataset (10,000 mixtures), and focuses on property prediction, rather than mixture representation learning. To work in the low-data regime of our olfactory mixture dataset, POMMIX uses pre-training techniques (Honda et al., 2019; Shoghi et al., 2023; Goh et al., 2018) and designed inductive biases to improve the expressivity of the molecular representation and attention mechanisms (Wang et al., 2019; Xiong et al., 2020; Maziarka et al., 2020).

## 2 METHODS

### 2.1 DATA

We combine mono-molecular datasets and multi-molecular (mixture) datasets. Mono-molecular datasets list a set of odor labels ("grassy", "fishy", etc.) for a single molecule, and the most exhaustive compilation is found in the GoodScents/Leffingwell (GS-LF) dataset (Barsainyan et al., 2024). We further clean the GS-LF dataset by canonicalizing SMILES (Weininger, 1988) strings with RDKIT (Landrum et al., 2022), removing duplicate entries, removing inorganic, charged or multi-molecular (e.g. salts) entries, removing molecules with molecular weight $< 20$ and $> 600$, and small inorganic molecules. We further removed infrequently applied odor labels that appeared for fewer than 20 molecules and subsequently removed molecules with no remaining labels (see Appendix A.1 for details on dataset cleaning).

Multi-molecular datasets were compiled from previous publications, hereby referred to as **Snitz** (Snitz et al., 2013) (containing data from Weiss et al. (2012)), **Ravia** (Ravia et al., 2020), and **Bushdid** (Bushdid et al., 2014). Data for each of these publications was obtained from *pyrfume* (Castro et al., 2022). In aggregate, we have 743 unique mixtures, containing between 1 to 43 unique molecular components (Figure 2a).

These mixtures are described by 865 pairwise mixture comparisons (Figure 2b) corresponding to labels from two types of experiments:

- **Explicit similarity (Snitz, Ravia)**: Participants are asked to explicitly rate the perceptual similarity of two mixtures from 0 (completely similar) to 1 (completely different). The final similarity for a mixture pair is averaged across all participants.
- **Triangle discrimination (Bushdid)**: Participants are provided three mixtures, of which two are identical, and asked to identify which mixture was different. These results are aggregated for each mixture triplet, and the percentage of correct identifications is treated as the label for the two unique mixtures in the triplet.

We note that the interpretation of the triangle discrimination task is congruent with the explicit similarity task, as a score of "1.0" in a triangle discrimination task shows that all tested participants could identify the mixture that was different, which meant that the two unique mixtures in the triplet were very perceptually distinct. Thus, in an explicit similarity test, this pair of mixtures would also have a score of "1.0". While these two tests are theoretically equivalent in their extremes (0 = perfect discrimination, 1 = equal to chance), calibration of intermediate scores may differ. We did not attempt to correct for this effect.

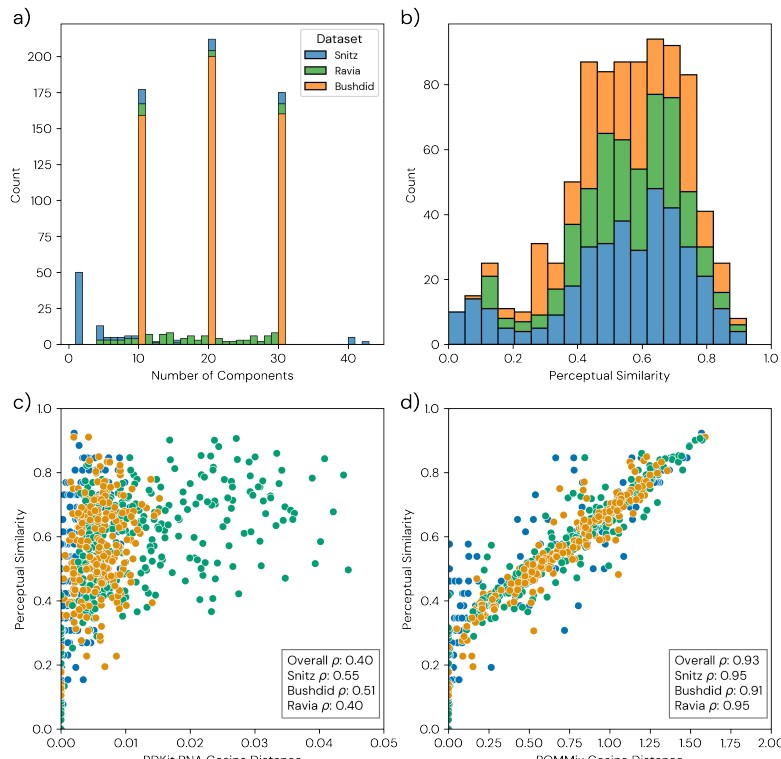

Figure 2: **Snitz, Ravia, and Bushdid mixture datasets at a glance**. **a**) Most mixtures contain 4-30 molecules, with a handful of single-molecule data as a measurement baseline. **b**) Most mixtures are somewhat different (0.4-0.8 averaged human response), with a smaller number of outright dissimilar measurements. **c**) Standard RDKIT cheminformatics molecule features, aggregated across the mixture with mean, standard deviation, minimum, and maximum (as described in Soelch et al. (2019); Corso et al. (2020)) correlate poorly with perceptual similarity, while **d**) POMMIX embeddings are carefully tuned for the task of discriminating mixture percepts. Pearson $\rho$ correlation constants are annotated in inset. Across all four subplots, color labels indicate the dataset source.

Intensity-balancing is a subtle part of mixture preparation. The naïve approach to preparing mixtures would be to use an equimolar or equivolume blend of components, but this approach tends to produce mixtures that are dominated by their most potent component. **Snitz**, **Ravia**, and **Bushdid** are instead intensity balanced, meaning that their components are first diluted to equal odor intensities using an odorless solvent (often, water or propylene glycol), and then mixed in equivolume proportions. POMMIX does not explicitly account for intensity or concentration of odorant mixtures, and would likely underperform in predicting mixture similarity if presented with mixtures that are not intensity-balanced.

## 2.2 MODELING

A schematic of the POMMIX model is provided in Figure 3. The POMMIX model can be divided into three hierarchical components: (1) a mono-molecular GNN POM embedding model, (2) a multi-molecular CHEMIX mixture attention model, and (3) a similarity scoring function.

The POM is a GNN which takes in molecular graphs derived from the SMILES representations of molecules. Each graph, written as $G = (U, V, E)$, consists of a special global vertex $U$ connected to all other vertices $V$, and a set of edges $E$. The global vertex $U$ encodes overall properties of the molecule and is initialized with 200 normalized RDKIT cheminformatics molecular descriptors (Landrum et al., 2022) obtained from DESCRIPTASTORUS (Kelley et al., 2024). The atoms of the molecules are the vertices (nodes), with node vectors $V = \{v_i\}_{i=1}^{N_v}$ for a molecule with $N_v$ atoms, where $v_i$ are 85-dimensional feature vectors encoding atomic properties. Covalent bonds between

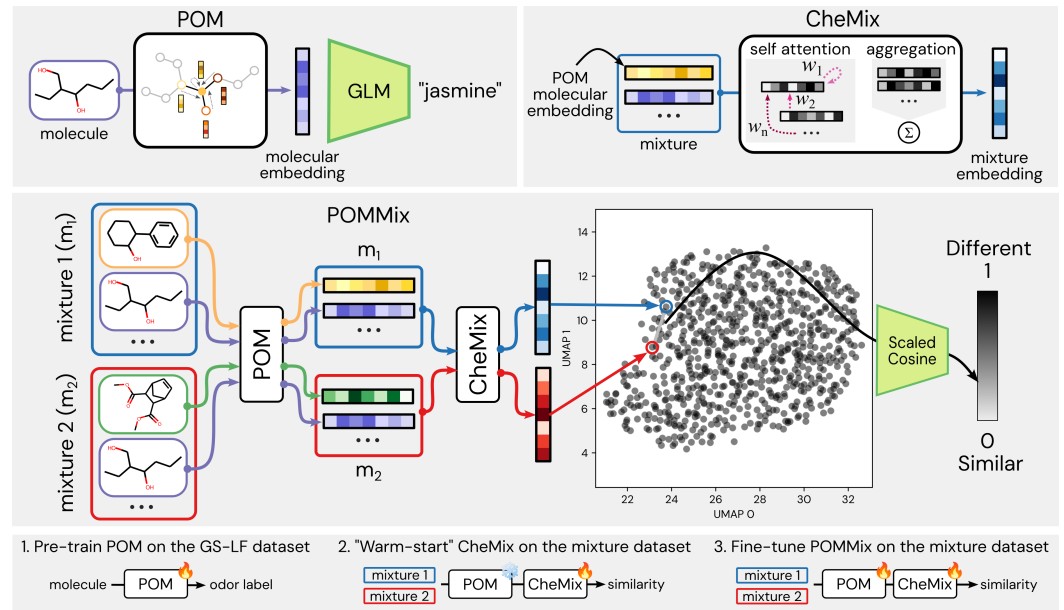

Figure 3: **The POMMIX model combines POM with mixture modeling**. (*Top*) The POM model with a generalized linear model (GLM) is pre-trained with mono-molecular olfactory data, and mixture modeling is performed through the CHEMIX attention model. (*Middle*) The two modules are joined to produce mixture embeddings which are trained to encode the olfactory perceptual distance of two mixtures using a scaled cosine distance predictor head. (*Bottom*) A multi-step model fitting procedure is used, where certain model weights are updated (flame) and other pre-trained model weights are frozen (snowflake).

atoms are represented as edges $E = \{(e_k, r_k, s_k)\}_{k=1}^{N_e}$ for a molecule with $N_e$ bonds, where $e_k$ stores a 14-dimensional feature vector of edge properties, and $r_k, s_k \in [1, \ldots, N_v]$ are indices that indicate the two atoms that the bond joins together. Note $r_k \neq s_k$, since bonds must be between two different atoms (see Appendix A.2 for detailed descriptions of node and edge properties).

The POM GNN uses the GRAPHNETS architecture (Battaglia et al., 2018), with message-passing blocks for the edge, node and global properties of the molecular graphs. The architecture is designed to be lightweight in order to avoid overfitting on the limited amounts of olfactory mixture data. Edge updates use feature-wise linear modulation (FiLM) layers (Perez et al., 2017; Brockschmidt, 2019), while node updates use graph attention layers (Veličković et al., 2017; Brody et al., 2021) with self-attention. The global embeddings are updated through principal neighborhood aggregation (PNA) (Corso et al., 2020; Zaheer et al., 2017). The GNN is composed of four of these GRAPHNET layers, and the final global embedding serves as the POM embedding.

The CHEMIX model processes a set of molecular POM embeddings, and generates an embedding representing the entire mixture. Mixtures are first represented by concatenating POM embeddings of constituent molecules, and mixtures with fewer molecules are padded to the length of the largest mixture. CHEMIX uses molecule-wise self-attention, where each molecule attends to all other molecules, followed by PNA. This ensures invariance of the mixture embeddings in the permutation of molecules within a given mixture. This model can be viewed as isomorphic to a GAN on a fully-connected graph (Joshi, 2020), with each molecule as a node, and the mixture embedding is the global embedding.

Finally, the distances between the mixture embeddings are obtained through a similarity score. For this, we use a predictive head based on cosine distance (Koch et al., 2015), commonly used for distance-aware high-dimensional learned representations. A final two-parameter linear layer is used to encode for human bias and experimental noise present in the dataset (see section 3), followed by a HardTanh activation to enforce output in the [0,1] range, while maintaining linearity. We note that this scaled cosine prediction head is invariant to the order of the mixtures due to the symmetry of the cosine distance operation.

## 2.3 TRAINING AND OPTIMIZATION

In order to effectively train a deep learning model in a low-data regime, we adopt a transfer learning strategy (Figure 3). The POM GNN is first pre-trained to predict the olfactory binary multi-labels of molecules with binary cross-entropy loss on the GS-LF dataset, using a 80/20 training/validation random split. All training is performed using the Adam optimizer (Kingma & Ba, 2014). To determine the architecture, we perform a Bayesian optimization hyperparameter search to maximize the area under receiver operator curve (AUROC) metric. Early stopping is used to prevent overfitting. The best model achieves a validation AUROC 0.884, in line with previous work (Sanchez-Lengeling et al., 2019). We explore other graph models such as GRAPHORMER (Shi et al., 2022; Ying et al., 2021) and GPS (Rampášek et al., 2022), but we find the GRAPHNETS architecture to be competitive with the modern state-of-the-art graph models for our dataset (Table A2).

The frozen POM embeddings from the pre-trained GNN form the vector representation of mixtures for the CHEMIX model. Again, the architecture is determined through hyperparameter tuning. The training is performed with mean absolute error (MAE) loss on a 80/20 training/validation split of the combined mixture dataset, stratified across **Snitz**, **Ravia**, and **Bushdid**. The stratification process fixes the proportion of each dataset across the splits, ensuring equal representation of any experimental differences. To avoid vanishing gradients due to the HardTanh activation, the linear model in the scaled cosine distance prediction head is initialized with bias $b = 0.5$, and the slope is clamped to ensure $m > 0$ and maintain the directionality of the cosine distance. Additionally, we ablate the CHEMIX prediction head, and find that the scaled cosine prediction head is optimal for learning mixture embedding similarities (Table A3). Early stopping terminates on maximal validation Pearson correlation coefficient ($\rho$) between the ground truth labels and the prediction. The optimal model found in the search achieved a maximal $\rho = 0.794$ on the validation set. Further details about the hyperparameter tuning for both models are provided in Appendix A.3.

In the final stage of training POMMIX, the POM GNN is directly joined to the CHEMIX model, and all model weights are allowed to vary. A lower learning rate is used for the POM GNN model weights, as they are already well-conditioned from pre-training on the larger mono-molecular odor dataset. The results following this section are based on the final POMMIX model. Models were built with PYTORCH (Paszke et al., 2019) and PYTORCH GEOMETRIC (Fey & Lenssen, 2019).

## 3 RESULTS

We evaluate our approach on the mixture dataset by training and testing on 5-fold cross-validation (CV) splits, stratified across the **Snitz**, **Ravia**, and **Bushdid** datasets. For early stopping, a validation split is randomly split from the training set, producing a final split of 70/10/20 training/validation/test sets. The performances of the models are then evaluated on the test sets.

We evaluate POMMIX on a progressive ladder of modeling components. For the simplest baseline, we follow the methods of Snitz et al. (2013), who performed extensive feature selection on molecular descriptors, which are then averaged together for the mixtures (see Appendix A.4). The angle distance between the vector descriptors are then correlated with the experimental results. We perform the same analysis using normalized RDKIT molecular features on our aggregated mixture dataset. We ensure that the feature selection is only performed on the training set.

We also provide comparisons with the gradient-boosted random forest XGBOOST model (Chen & Guestrin, 2016), and use features with varying levels of inductive biases (further details in section A.5). Mixture representations are created with PNA-style aggregation of molecular descriptors, including RDKIT features, or the frozen POM embeddings. Additionally, we augment the training data by permuting the mixture pairs, as the symmetry of the mixture similarity is not encoded in XGBOOST.

## 3.1 PREDICTIVE PERFORMANCE

We report results across three metrics: Pearson correlation coefficient $\rho$, root-mean-squared error (RMSE), and the Kendall ranking coefficient $\tau$, each reflecting different strengths of the model. The test results compiled from the CV splits for all models evaluated are shown in Figure 4, with metrics tabulated in Table 1. We show that incorporating more inductive biases into the model leads

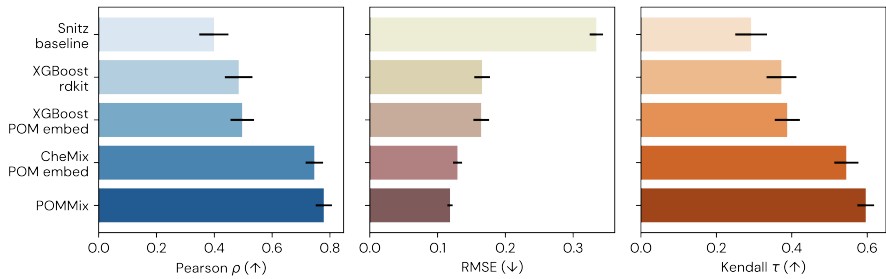

Figure 4: **Model performances on mixture dataset**. Pearson $\rho$, RMSE, and Kendall $\tau$ for all baselines and models evaluated. Model complexity increases from top to bottom. Parity plots available in Appendix A.6.

to a dramatic increase in model performance. The Snitz baseline of angle distances calculated from empirically selected features produces a weak positive correlation with the ground truth distances, but has high RMSE when treated as a regression problem. The XGBOOST model improves upon these predictions, and explicitly models the experimental distances, achieving significantly lower RMSE than the Snitz baseline. However, the correlation and ranking of the mixture similarity is only slightly increased through the use of the boosted RF model. When applying the XGBOOST model to the POM embeddings, we find that the performance is only slightly improved compared to the RDKIT descriptors, signaling that deep learning architectures are needed to extract useful information out of the POM embeddings.

For our approaches, CHEMIX shows excellent test performance for predicting olfactory mixture similarities, even when trained with frozen POM embeddings, demonstrating the efficacy of incorporating domain knowledge and inductive biases into model architectures. For POMMIX, we observe further increases in model performance when the POM and CHEMIX are trained end-to-end, further fine-tuning the POM embeddings for use in mixture representations. We note that the end-to-end training results in larger improvements in Kendall $\tau$ than in $\rho$. We hypothesize that inherent human noise in the experimental results create a performance ceiling for the model's real-valued predictive capabilities. However, the ranking correlation still improves as it is more robust to experimental noise and outliers (Tom et al., 2024). We further explore this human bias in Section 3.3. Finally, we note that our attempts to augment the dataset with pairs of mono-molecules labeled by their GS-LF odor label Jaccard distances led to modest improvements for the CHEMIX model, but showed no improvements for POMMIX (see Appendix A.8 and Table A5 for details on data augmentation).

Table 1: **Model performances on mixture dataset**. 5-fold cross validation metrics for baseline models, CHEMIX and POMMIX. The mean and standard deviation are reported. Other ablated models are provided in Table A4.

| Model | Test predictive performance | | |
| --- | --- | --- | --- |
| | Pearson $\rho$ ($\uparrow$) | RMSE ($\downarrow$) | Kendall $\tau$ ($\uparrow$) |
| Snitz Baseline | $0.399 \pm 0.050$ | $0.334 \pm 0.010$ | $0.292 \pm 0.042$ |
| XGBOOST + RDKIT | $0.485 \pm 0.048$ | $0.166 \pm 0.012$ | $0.373 \pm 0.040$ |
| XGBOOST + POM | $0.497 \pm 0.041$ | $0.165 \pm 0.012$ | $0.388 \pm 0.033$ |
| CHEMIX + POM | $0.746 \pm 0.030$ | $0.130 \pm 0.007$ | $0.545 \pm 0.032$ |
| POMMIX | $\mathbf{0.779 \pm 0.028}$ | $\mathbf{0.118 \pm 0.004}$ | $\mathbf{0.596 \pm 0.022}$ |

## 3.2 GENERALIZATION TO NEW MIXTURE SIZES AND MOLECULES

We further study the effects of the inductive biases of the model, and the capabilities of POMMIX in explaining physical olfactory phenomena. In particular, we study the generalization of POMMIX to different splits based on the number of mixture components, and the molecular identities within the mixtures.

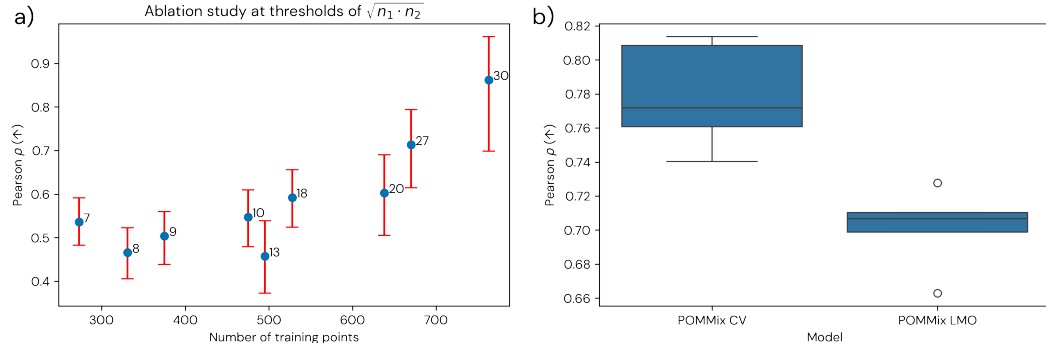

Figure 5: **Generalization to new mixture sizes and molecules**. **a)** Ablation study with training data only containing mixtures with geometric average number of molecules less than a threshold. The thresholds are indicated for each split. **b)** Boxplot of POMMIX test Pearson correlation on random CV splits, and the LMO splits.

In Figure 5a, we show the test results of an ablation study, in which the training data is ablated based on thresholds on the geometric average number of components found in a mixture. In other words, for a given threshold, the training set only contains mixtures with components that have a geometric mean number of components less than the threshold, and the test set contains only mixtures that are above the threshold. We observe sufficient generalization capabilities of the model to larger mixtures, achieving performances similar to the RDKIT baselines, even when the training sets are thresholded at ten mixture components, and only about two-thirds of the available training data. We also observe a general increase in test performance, measured by $\rho$, as the training set grows, indicating that more high quality experimental olfactory mixture data can greatly improve the modeling performance.

We observe a significant decrease in performance when considering new chemistries. For Figure 5b, we study POMMIX performance on leave-molecules-out (LMO) splits, in which the test sets are split from the dataset such that certain molecules will not appear in the training set. Note that, unlike the random CV splits, the training sets are not mutually exclusive, since there is significant overlap in the molecular identities across different mixtures. This additional challenge in studying new molecules is an important consideration when validating models, and also planning future mixture similarity experiments. More olfactory mixture data with diverse arrays of molecules can help build better and more generalizable POMMIX representations.

### 3.3 EXPLORING OLFACTORY PHENOMENA WITH POMMIX EMBEDDINGS

The *white noise hypothesis* states that intensity-balanced mixtures with an increasingly large number of components become increasingly indistinguishable, even if they share no common molecular components, and approach a scent characterized as an "olfactory white" (Weiss et al., 2012). Using the POMMIX embedding, we reproduce the "olfactory white" phenomena (Figure 6a). In our investigation, we observe this decrease in POMMIX embedding distances as a function of the geometric mean of components in mixture pairs for our larger dataset, which includes **Bushdid** and **Ravia**. This demonstrates the ability of POMMIX in capturing and explaining physiological olfaction phenomena, allowing it to build toward an expressive odor perceptual space.

It is important to note that the perceptual similarity metrics obtained across the datasets are inherently subjective and biased as they are gathered from humans. We show a subset of the data where the panelists are asked to rate the similarities of two *identical* mixtures, and show that a significant portion of identical mixture pairs (60 out of 63) are labeled as having non-zero similarities (Figure 6b). While the observed bias could be descriptive of average human olfactory inaccuracies, because the number of panelists sampled was low ($\sim$300), the bias could be local to the panelists. This human bias is modeled by the learned bias term of the scaled cosine similarity prediction head. In general, we observe that the learned bias is slightly higher than the dataset bias. When we physi-

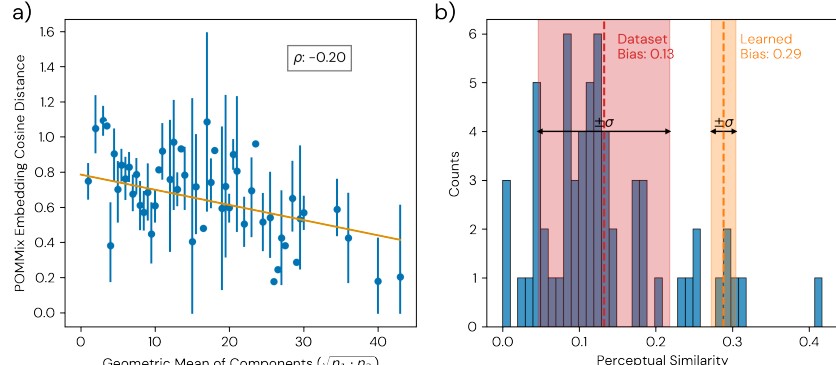

Figure 6: **Investigating inductive biases in perceptual olfactory data**. **a)** The white noise hypothesis (Weiss et al., 2012), where larger mixtures become less distinguishable from one another. **b)** Inherent human inaccuracies in the identification of two identical mixtures (data from **Snitz**). The mean (dashed line) and standard deviation (shaded region) of perceptual similarity for identical mixtures within the dataset is in red, while the learned bias of the similarity prediction head is in orange. The learned biases are averaged across the five CV splits.

cally ground the model by enforcing that the learned similarity of identical mixtures should be zero (i.e., $b = 0$), we observe poorer model performance (see Appendix A.6). This could be a result of having a small dataset (< 1,000 points); enforcing this inductive bias may be more relevant when considering larger data regimes and experiments with more human panelists, where the perceptual distance of identical mixtures approach zero, making the model more generalizable and not subject to the bias of a specific dataset.

### 3.4 BUILDING INTERPRETATIONS OF MIXTURES

An unanswered question relevant to mixture modeling is how the mixture components interact with each other and contribute to the prediction of mixture similarities. To probe at this question and generate hypotheses for future investigation, we modified CHEMIX to be more interpretable as an additive model (Agarwal et al., 2021). Specifically we express the self-attention-based mixing component as a one-layer additive model by using sigmoid normalization (Ramapuram et al., 2024) rather than softmax, allowing the model to attend to all components, and forcing the value vectors to be positive via a ReLU activation. In a simplified way, this is a pairwise interaction model. Although this modified model is simpler and more constrained, it achieves performance comparable to that of our best model.

In Figure 7, we showcase how such sigmoidal self-attention maps can be used to analyze the information passing between representations of molecules within a mixture. More complex examples can be found in Appendix A.9. In this simple example, when comparing the GS-LF labels associated to each molecule (Figure 7a) to the attention weights attributed to each query (Figure 7b), we notice that the strongest "interaction"—namely, the highest attributed attention weight—is found between query molecule 1 and key molecule 3. In general, we observe that molecules that are most different from the rest, either by chemical structure (e.g., presence of N or S atom) or by olfactory perception (e.g., presence of rare or numerous labels), tend to have stronger interactions. To further our analysis, we derive label-guided structural heuristics about molecules across the set of unique mixtures in Appendix A.10. We find that higher attention is attributed to chemical structures with strong, pungent, and unique smells. These include compounds with sulfur, nitrogen, and aromatic structures. However, it is important to keep in mind that the attention map showcased here is intrinsically linked to the task of differentiating between mixtures and is therefore likely biased towards attributing higher attention weights to molecular embeddings that carry discriminative power only relative to this task. Careful experimentation with synthetic mixture tasks and dataset, where the number of data points is not as limited, might provide guidance on the strengths and failure modes of these approaches to interpretability (Sanchez-Lengeling et al., 2020). Prospective validation from new experimentation would also strengthen these hypotheses.

A multi-headed ($k$) softmax attention mechanism can be interpreted as attending to $k$ tokens (Joshi, 2020; Sanchez-Lengeling et al., 2021). In understanding interpretability, a natural question for mixtures might be: how many compounds do we need to attend to on average? Figure 7c attempts to answer this by looking at the average number of interactions per compounds across the dataset. We consider three attention weight cutoffs (0.3, 0.4 and 0.5) to define a significant interaction and observe that the number of interactions grows approximately linearly with the number of components in the mixtures. For the 0.5 cutoff, this is approximately two interactions per compound for mixtures of less than 30 components. We observe an increase in average number of interactions after 30 components; however, we also note that data is quite sparse here, precluding the formation of firm conclusions about the data (Figure 2a).

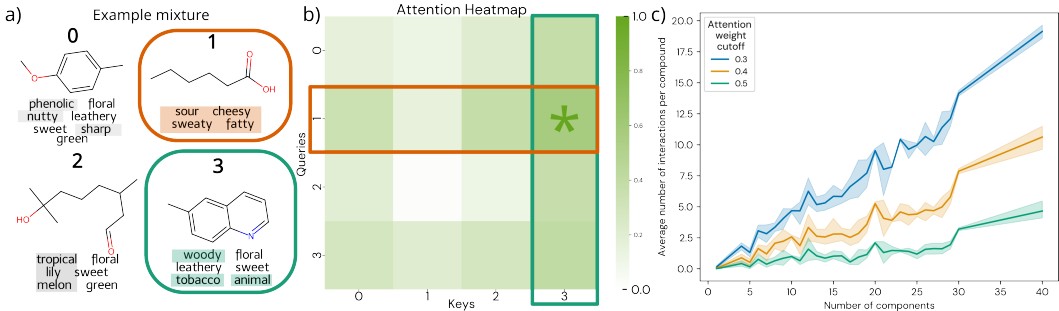

Figure 7: **Mixture attention maps**. **a)** Example mixture with molecules and their odor labels. **b)** Sigmoidal self-attention heatmap, with compound 1 and 3 highlighted. Strongest interaction is indicated with an asterisk. **c)** Number of average interactions per compound as function of mixture size across all datasets. Each color represents a different threshold for a meaningful interaction.

## 4 CONCLUSION AND DISCUSSION

We introduce POMMIX, an extension of the POM for predicting olfactory similarities between mixtures of molecules. Our approach combines graph neural networks for molecular representation, with attention mechanisms for mixture modeling, and incorporates inductive biases by considering cosine similarities between mixture embeddings to predict olfactory similarity. POMMIX demonstrates state-of-the-art performance, creating meaningful representations of olfactory mixtures, and we show how each component of inductive bias contributes to this performance.

Our work highlights the value of incorporating domain knowledge and inductive biases, particularly in low-data regimes. By respecting problem-space symmetries, we create a flexible and expressive representation for olfactory mixtures, offering a potential solution for modeling other multicomponent systems. Furthermore, we provide a method towards interpretable modeling of mixture components interactions studying the attention weights of mixture components in CHEMIX and studying how molecular information attends to itself within a mixture.

While POMMIX shows promising results, we acknowledge several limitations. The small size of the available mixture dataset (< 1,000 samples) raises concerns about overfitting, despite our regularization efforts. Additionally, the limited coverage of chemical odorant space in current public datasets (only ~200 unique odorant compounds) constrains the model's ability to generalize to a wider range of chemical compounds. We also observed challenges in generalizing to new datasets due to potential distribution shifts from varying experimental setups and human biases. Despite making conscious design choices on the modeling side, the interpretability of mixture components interactions in CHEMIX remains qualitative; further experimental investigations are required to validate our conjectures.

We believe that the primary bottleneck in advancing olfactory modeling is the generation of high-quality, diverse, and representative datasets. Future work should focus on expanding the coverage of chemical space, incorporating various experimental conditions (e.g., dilution, intensity), and collecting rich textual descriptions of odors. Such comprehensive datasets will be crucial for developing more robust, interpretable and generalizable models of olfactory perception.

## 5 REPRODUCIBILITY STATEMENT

We have made significant efforts to ensure our methodology can be replicated by other researchers. All data and code used are provided in https://anonymous.4open.science/r/anon-iclr2025-DE62. We use open-source software, including PYTORCH, PYTORCH GEOMETRIC, and RDKIT. Our manuscript details the model architecture, training procedures, and evaluation metrics. We outlined our data sources and preprocessing steps, including specific criteria for removing molecules and odor labels. Details on dataset cleaning are provided in Appendix A.1, model details are provided in Section 2.2 and Appendix A.2, and the training process and hyperparameter tuning are provided in Section 2.3 and Appendix A.3, respectively. Additionally, the splits used for all experiments are provided as well. We are committed to ensuring other researchers can build upon our findings and verify our results.

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

# A  APPENDIX

## A.1  GS-LF DATASET FILTERING

The open-source version of the GoodScents/Leffingwell (GS-LF) dataset by Barsainyan et al. (2024) initially contains 4983 molecules with 138 odor descriptors. These filters were applied in the following order:

- **Inorganic atom filter.** 110 molecules containing these atoms were removed: ["He", "Na", "Mg", "Al", "Si", "K", "Ca", "Ti", "V", "Cr", "Fe", "Co", "Cu", "Zn", "Bi"].
- **Duplicate SMILES filter.** 0 duplicate molecules were removed.
- **Salts and charged molecule filter**. 10 molecules containing charges (including salts) were removed.
- **Multimolecular filter.** 36 SMILES strings containing multiple molecules were removed (characterized by SMILES strings containing the "." character).
- **Molecular weight filter**. 1 molecule with MW < 20 was removed. 11 molecules with MW > 600 were removed.
- **Non-carbon molecule filter.** 1 molecule containing only non-carbon atoms was removed.

This filtering process results in a dataset of 4814 molecules, which was then used to train the POM.

## A.2  DETAILS OF MOLECULAR GRAPH REPRESENTATION

The node features used in the molecular graph representation as input to the POM GNN are 85-dimensional one-hot encoding vectors, encoding categorical information about the atoms. The edge features encode the categorical information about the bonds as 14-dimensional one-hot encoding vectors. The molecular information for the features are shown in Table A1.

Table A1: **Features for node and edge features of molecular graphs**. All categories are one-hot encoded and stacked to give a singular bit vector. UNK stands for "unknown", and is a catch-all category.

| Node features | Categories |
| --- | --- |
| Atomic number | 1 (hydrogen) to 54 (iodine), UNK |
| Atom degree | 0, 1, 2, 3, 4, 5, UNK |
| Formal charge | -2, -1, 0, 1, 2, UNK |
| Chirality | unspecified, CW, CCW, other, UNK |
| Number of hydrogens | 0, 1, 2, 3, 4, 5, 6, 7, 8, UNK |
| Hybridization | sp, sp2, sp3, sp3d, sp3d2, UNK |
| Aromatic | True/False |

| Edge features | Categories |
| --- | --- |
| Bond type | single, double, triple, aromatic, UNK |
| Is conjugated | True/False |
| In ring | True/False |
| Stereo-configuration | none, $Z$, $E$, $cis$, $trans$, any, UNK |

### A.3 HYPERPARAMETER SEARCH

We perform hyperparameter searches for the pre-training of both the POM GNN and the CHEMIX models. For the POM GNN, we use Optuna (Akiba et al., 2019), with the Tree-structured Parzen Estimator algorithm (Bergstra et al., 2011), with a budget of 200 runs. The final embedding space is fixed to 196 dimensions. The node GAT model and edge FiLM model is fixed to a single layer, while the global PNA model has 2 layers. The search space is defined as follows (bolded values are the optimal):

- Number of GRAPHNETS layers: [2, 3, **4**]
- Hidden dimensions for all models: [64, 128, 192, 256, **320**]
- Dropout rate: [0, 0.05, **0.1**, 0.15, 0.2, 0.25, 0.3, 0.35, 0.4, 0.45, 0.5]
- Learning rate: [1e-2, 5e-3, 1e-3, 5e-4, **1e-4**, 5e-5]

For CHEMIX, the search was performed using Weights & Biases (Biewald, 2020) with BOHB (Falkner et al., 2018) algorithm and a budget of 200 runs. Early stopping was implemented with patience set to 100 epochs The search space was defined as follows (bolded values are the optimal):

- Embedding dimension: [32, 64, **96**, 128]
- Number of MolecularAttention (self attention) layers: values: [0, **1**, 2, 3]
- Number of attention heads: values: [1, 4, **8**, 16]
- Addition of an MLP head on top of MolecularAttention: ["True", **"False"**]
- Type of molecular aggregation: ["mean", **"pna"**, "attention"]
- Scaled cosine activation function: ["sigmoid", **"hardtanh"**]
- Attention type: ["standard", **"sigmoidal"**]
- Dropout rate: [0, 0.05, **0.1**, 0.15, 0.2, 0.25, 0.3, 0.35, 0.4, 0.45, 0.5]
- Learning rate: [**8e-5**, 1e-4, 5e-4, 8e-4, 1e-3]
- Loss type: [**"mae"**, "mse", "huber"]

### A.4 PROCEDURE FOR SNITZ BASELINE

The Snitz baselines are reproduced following the procedure outlined in Snitz et al. (2013). There are three steps involved in optimizing the angle similarity model for the best descriptors. Prior to the optimization campaign, we normalize the 200 RDKIT features obtained from DESCRIPTASTORUS (Kelley et al., 2024) and average across all the molecules in the mixture.

In step 1, we determine the appropriate number of descriptors by randomly sampling 20,000 times, without replacement, $n \in [2, 200]$ descriptors, resulting in 199 sets of 20,000 samples each of predictions. We then evaluate the RMSE from the predictions of the similarity model for each value of $n$. The optimal number of descriptors was scored by minimizing $\mu_{RMSE} - \sigma_{RMSE}$ across all 20,000 samples for a given $n$. The appropriate number of descriptors was between $n = 5$ and $n = 7$, depending on the CV split.

In step 2, we evaluate the efficacy of each descriptor. We set the number of descriptors $n$ for each CV split based on step 1, and randomly sampled $n - 1$ descriptors 2,000 times. Then, cycling through each individual descriptor, we appended it to each set of sampled $n - 1$ descriptors, producing a vector of $n$ features, and again evaluated RMSE from the predictions of the similarity model. We take the mean RMSE from the 2,000 samples and the most relevant descriptors are determined by minimizing the mean RMSE.

In step 3, we first calculate the score of each descriptor from step 2. The 2,000 samples of $n - 1$ features with a specially appended feature $i \in [1, 200]$ provides a score for the $i$-th descriptor in the representation. This score for the $i$-th descriptor is given by

$$score(i) = \max\left(0, -\frac{\text{RMSE}_i - \mu_{\text{RMSE}}}{\sigma_{\text{RMSE}}}\right), \tag{1}$$

which only provides a positive score if the feature achieves lower RMSE than the average RMSE achieved over all features. Only positive scored features are kept. We then randomly sample, 4,000 times, $n = 5$ to $n = 7$ descriptors depending on the appropriate CV split out of the set of descriptors that performed better than the average RMSE value (i.e. positive score). Out of the 4,000 samples, we pick the best-performing set of descriptors (lowest RMSE) on the training set and perform a final evaluation on the test set. This procedure produces the values found in Section 3.

## A.5 XGBOOST MODELING

The XGBOOST model was given a maximum of 1,000 estimators and tree depth of 1,000. To ensure the model does not overfit, we use the validation set for early stopping, with a patience of 25 epochs. The model is trained with mean squared error, with a learning rate of 0.01.

## A.6 PARITY PLOTS FOR ALL BASELINE MODELS, CHEMIX, POMMIX, AND ZERO-BIAS POMMIX

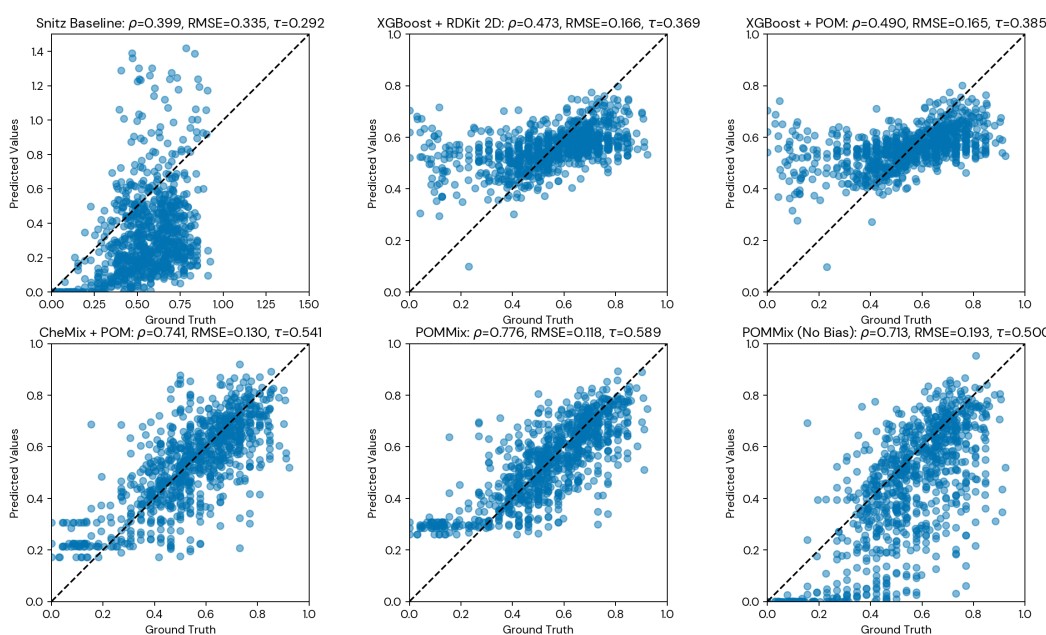

Figure A1: **Parity plots for all models evaluated**. Ground truth labels versus predicted values across all five cross-validation splits, with Pearson $\rho$, RMSE and Kendall $\tau$ reported.

## A.7 ADDITIONAL ABLATION STUDIES

In addition to the ablation studies in Section 3, we perform additional ablations of the POM graph model, the molecular featurization, and the CHEMIX prediction head.

In Table A2, we compare the chosen GRAPHNETS GNN with graph transformer models GRAPHORMER (Shi et al., 2022; Ying et al., 2021) and GPS (Rampášek et al., 2022). These models have shown state-of-the-art performance on large molecular datasets, such as Open Graph Benchmark (OGB) (Hu et al., 2020; 2021), the Open Catalyst Challenge (Chanussot et al., 2021), and the ZINC 250k dataset (Gómez-Bombarelli et al., 2018).

Table A2: **Other GNN models**. Validation results on GS-LF model for Graphormer and GPS models. While both graph transformers achieve state-of-the-art performances on larger molecular datasets, the lightweight and tuned POM performs as well as or better than the models.

| Model | Validation AUROC ($\uparrow$) |
| --- | --- |
| GRAPHORMER (slim) | 0.8564 |
| GPS | 0.8647 |
| GRAPHNETS POM | **0.8843** |

In Table A3, we provide the cross validation test performance results for CHEMIX with frozen POM embeddings and different prediction heads. We train four additional models of CHEMIX with

different prediction heads: Mean + Linear, Concatenate + Linear, PNA-like + Linear, and unscaled cosine distance. The unscaled cosine distance prediction head achieves poor RMSE, but better correlation metrics ($\rho$ and $\tau$) when compared to the regressive prediction heads. Of the aggregation methods, we find that the PNA-like aggregation produces the best results with the linear regression head. The scaled cosine prediction head combines the strengths of both, achieving the best test performance across all three metrics. Additionally, we want to imbue the mixture embedding space with a notion of distance and similarity. The scaled cosine similarity was finally chosen, based on our experiments, as the best POMMIX prediction head.

Table A3: **Ablation of CHEMIX prediction head**. 5-fold cross validation metrics for CHEMIX with various prediction heads. The mean and standard deviation are reported. The scaled cosine distance is what was finally chosen for POMMIX, and is reproduced here for comparison.

| | Test predictive performance | | |
|---|---|---|---|
| CHEMIX prediction head | Pearson $\rho$ ($\uparrow$) | RMSE ($\downarrow$) | Kendall $\tau$ ($\uparrow$) |
| Mean + Linear | $0.085 \pm 0.123$ | $0.204 \pm 0.018$ | $0.097 \pm 0.057$ |
| Concatenate + Linear | $0.266 \pm 0.089$ | $0.201 \pm 0.009$ | $0.190 \pm 0.063$ |
| PNA-like + Linear | $0.477 \pm 0.066$ | $0.174 \pm 0.005$ | $0.295 \pm 0.048$ |
| Cosine distance | $0.676 \pm 0.049$ | $0.208 \pm 0.009$ | $0.436 \pm 0.028$ |
| Scaled cosine distance | $\mathbf{0.746 \pm 0.030}$ | $\mathbf{0.130 \pm 0.007}$ | $\mathbf{0.545 \pm 0.032}$ |

In addition to RDKIT and POM embeddings, we also use the MOLT5 (Edwards et al., 2022) chemical language model embeddings. MOLT5 uses self-supervised training to build a transformer model trained on unlabeled natural language and molecular strings, and is then fine-tuned on annotated chemical data. We use these embeddings with the XGBOOST baseline, and also the CHEMIX model. These models give test performance metrics (Table A4) that are worse than the RDKIT molecular descriptors for the respective models. Across all models, we find the best performance with the POM embeddings, which were fine-tuned for our final POMMIX model. Recent work by Shin et al. (2018) studying the use of transformer-based language models and in combination with graph models show that GNN methods are still optimal for this modeling problem.

Table A4: **Model performances on mixture data with additional ablation of features**. 5-fold cross validation metrics for all baseline models, CHEMIX and POMMIX. The mean and standard deviation are reported. We include additional results (underlined) with MOLT5 chemical language model embeddings and RDKIT features. Results from Table 1 are reproduced here for comparison.

| | Test predictive performance | | |
|---|---|---|---|
| Model | Pearson $\rho$ ($\uparrow$) | RMSE ($\downarrow$) | Kendall $\tau$ ($\uparrow$) |
| Snitz Baseline | $0.399 \pm 0.050$ | $0.334 \pm 0.010$ | $0.292 \pm 0.042$ |
| XGBOOST + MOLT5 | $0.432 \pm 0.030$ | $0.171 \pm 0.012$ | $0.347 \pm 0.036$ |
| XGBOOST + RDKIT | $0.485 \pm 0.048$ | $0.166 \pm 0.012$ | $0.373 \pm 0.040$ |
| XGBOOST + POM | $0.497 \pm 0.041$ | $0.165 \pm 0.012$ | $0.388 \pm 0.033$ |
| CHEMIX + MOLT5 | $0.672 \pm 0.021$ | $0.144 \pm 0.006$ | $0.498 \pm 0.023$ |
| CHEMIX + RDKIT | $0.732 \pm 0.030$ | $0.132 \pm 0.007$ | $0.552 \pm 0.040$ |
| CHEMIX + POM | $0.746 \pm 0.030$ | $0.130 \pm 0.007$ | $0.545 \pm 0.032$ |
| POMMIX | $\mathbf{0.779 \pm 0.028}$ | $\mathbf{0.118 \pm 0.004}$ | $\mathbf{0.596 \pm 0.022}$ |

## A.8 PRE-TRAINING WITH AUGMENTED DATA

Due to the scarcity of mixture data, especially those with perceptual similarities between single molecules, we sought to investigate if we could augment data using available larger mono-molecular datasets. We investigated if the Jaccard distance between the odor descriptors of two molecules (obtained from GS-LF) was a good proxy to pairwise single-molecule perceptual similarities. Based on 75 single-molecular pairwise perceptual similarity measurements already in our dataset, we discovered a modest correlation ($\sim$0.49) between the Jaccard distance and the perceptual similarity

(Figure A2a). Thus, we pre-trained CHEMIX with this augmentation strategy with a total of 15571 augmented datapoints, followed by fine-tune training of POMMIX , but found that it did not provide improved structure for the embedding space for single-molecular mixtures (Figure A2b), where the pairwise distance of the POMMIX mono-molecular mixture embeddings remained the same. Additionally, the pre-training causes reduced model performance in all tracked metrics for CHEMIX (Table A5).

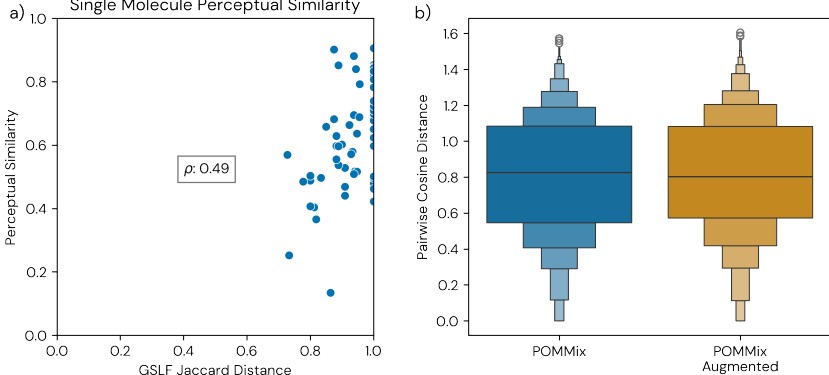

Figure A2: **Augmentation with GS-LF odor label Jaccard similarities**. **a**) Correlation between the Jaccard distance of the GS-LF odor labels of two single molecules, versus their perceptual similarity. **b**) Boxen plot of all pairwise cosine distances between the embeddings of single molecules for POMMIX, with and without augmentation.

Table A5: **Model performances on pre-training with augmented data**. 5-fold cross validation metrics for CHEMIX pre-trained with and without augmented data. The mean and standard deviation are reported.

| | Test predictive performance | | |
| --- | --- | --- | --- |
| Model | Pearson $\rho$ ($\uparrow$) | RMSE ($\downarrow$) | Kendall $\tau$ ($\uparrow$) |
| CHEMIX + POM (augmented) | $0.628 \pm 0.048$ | $0.163 \pm 0.007$ | $0.479 \pm 0.039$ |
| CHEMIX + POM | $\mathbf{0.746 \pm 0.030}$ | $\mathbf{0.130 \pm 0.007}$ | $\mathbf{0.545 \pm 0.032}$ |

## A.9 ATTENTION HEATMAP EXAMPLES

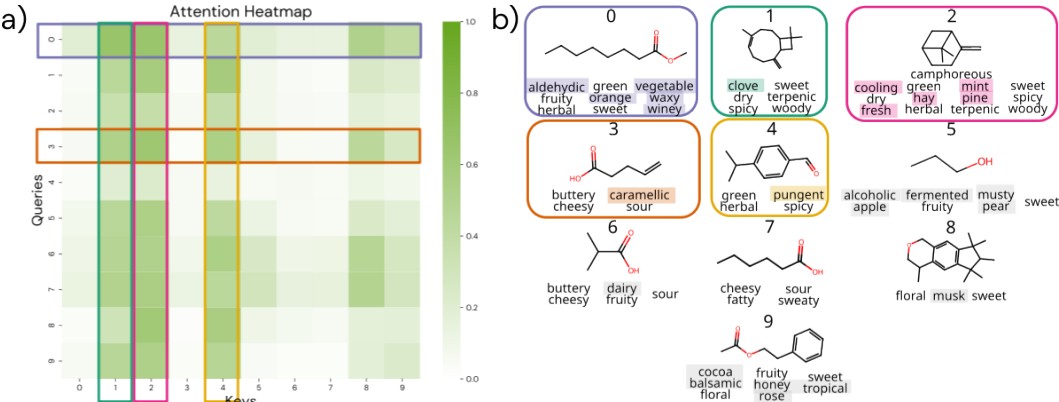

Figure A3: **Mixture attention map example with 10 components**. **a)** Sigmoid attention heatmap, compounds with 3 or more significant interactions (cutoff=0.5) are highlighted. **b)** Example mixture with molecules and their odor labels. Most interacting molecules are highlighted, and unique labels have a shaded rectangle.

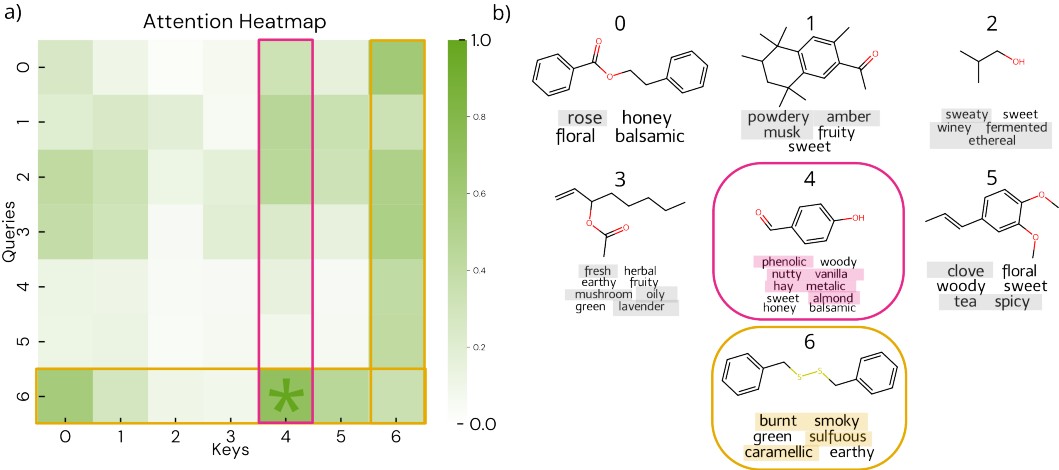

Figure A4: **Mixture attention map example with 7 components**. **a)** Sigmoid attention heatmap, compounds with 3 or more significant interactions (cutoff=0.5) are highlighted. Strongest interaction is indicated with an asterisk. **b)** Example mixture with molecules and their odor labels. Most interacting molecules are highlighted and unique labels have a shaded rectangle.

### A.10 MIXTURE SET LABEL-GUIDED STRUCTURAL INSIGHTS ON KEY MOLECULES

To derive structural heuristics across the entire set of unique mixtures, we analyze the key-ed molecules associated with extrema attention weight values for each query, focusing on queries that interact "strongly" with a key (an interaction is considered strong if the attention weight is above 0.5). We visualize the UMAP (McInnes et al., 2018) of the POM embeddings projected by CHEMIX through one linear layer of key-ed molecules exclusively found as maximizing/minimizing attention weights (Figure A5, left). We observe a clear separation between the two classes, suggesting that certain types of molecules are prioritized (high interactions) or de-prioritized (low interactions) when it comes to updating the molecular embeddings within a mixture.

We then performed hierarchical clustering (Müllner, 2011) of these key-ed molecules with SCIPY (Virtanen et al., 2020) based on the pairwise Jaccard similarity of the binary GS-LF odor descriptor labels. We selected a few representative molecules for each of the clusters and observed strong structural differences between them (Figure A5, right). This is implicitly expected from the structure-property relationship between scent and molecular structure. More importantly, we note that molecules within clusters are generally either "strongly" or "weakly" interacting, suggesting our model established a relationship between specific molecular structures and attention weight values. Through this analysis, we observe that ester/aldehydes with long alkane chains tend to have low interaction keys (cluster 2, 3 and 7; Figure A5, right), while sulfur-containing molecules and molecules containing aromatic rings tend to be highly interacting ones (cluster 1, 4 and 5; Figure A5, right). These structural insights derived from label-driven clustering confirm the idea that certain molecules receive more attention than others.

One possible explanation for why ester/aldehydes with long alkane chains are "non-interacting" keys would be that such molecules generally have a pleasant, sweet, or fruity smell. These odor descriptions are highly prevalent in the mixture dataset (119 (58.62%) "sweet" molecules and 92 (45.32%) "fruity" molecules in 203 unique molecules across the 743 unique mixtures), and could therefore not be informative in distinguishing mixtures. On the other hand, sulfur-containing molecules generally have a pungent, garlicky smell and occur less in the mixture dataset dataset (10 (4.92%) "garlic" molecules).

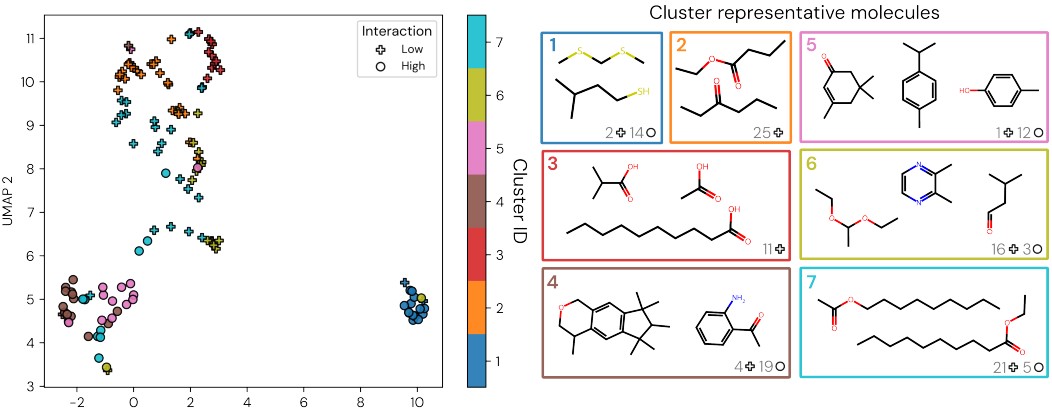

Figure A5: **Visualizing the embeddings of maximally/minimally interacting key-ed molecules across unique mixtures**. *(Left)* UMAP visualization of the embeddings of key-ed molecules exclusively found as maximizing/minimizing attention weights, for each query exhibiting significant interaction (attention weight > 0.5) across all unique mixtures. The interaction strength, determined by the attention weight, of the molecules are indicated by the markers. The molecules are colored by cluster identity. Molecules without GS-LF labels are excluded from the visualization. *(Right)* Representative molecules for each of the label-based Jaccard distance clusters. The number of strong/weak interaction molecules for each cluster is indicated in the bottom right corner of each box.

