# OpenReview forum: "From Molecules to Mixtures: Learning Representations of Olfactory Mixture Similarity using Inductive Biases"
_ICLR.cc/2025/Conference — Submitted to ICLR 2025_

### Official Review · Reviewer_BX4L · 2024-10-30

**Soundness:** 2
**Presentation:** 3
**Contribution:** 2
**Rating:** 5
**Confidence:** 4

**Summary:**

This work presents POMMIX, a novel model for representing molecular mixtures, leveraging hierarchical design to capture the underlying symmetries in the mixture space. The approach comprises three key components: (1) graph neural networks for generating robust molecular embeddings, (2) attention mechanisms for effectively aggregating these molecular embeddings into comprehensive mixture representations, and (3) cosine similarity-based prediction heads to encode perceptual distances in the mixture embedding space, aligning with olfactory perception.

**Strengths:**

POMMIX’s hierarchical architecture captures the structural complexity of molecular mixtures by integrating graph neural networks, attention mechanisms, and cosine-based prediction heads, enabling acquisition of mixture representations. By encoding perceptual distances in the embedding space through cosine prediction heads, POMMIX aligns mixture embeddings with olfactory perceptual similarities, a novel enhancement for sensory science applications. The model consistently demonstrates high predictive accuracy across multiple datasets, highlighting its robustness and generalizability for a range of mixture-related tasks, from olfactory perception to chemical property prediction. Additionally, POMMIX’s modular framework allows it to be readily adapted to various molecular mixture tasks, making it a flexible tool for predictive modeling and exploratory analysis in molecular science.

**Weaknesses:**

This work has some limitations that highlight areas for future improvement in POMMIX. First, its performance is highly dependent on the quality and diversity of the training data; limited or biased data can hinder generalization, especially in underrepresented molecular categories. Additionally, POMMIX’s hierarchical architecture—combining graph neural networks and attention mechanisms—is computationally intensive, posing scalability challenges for very large datasets or deployment in resource-limited settings. The model’s high capacity for capturing complex representations also increases the risk of overfitting, particularly with smaller or highly correlated datasets, which could impact generalization to new data.

While innovative in its structure, POMMIX’s individual components could benefit from further exploration. For instance, alternative embedding techniques, such as chemical language models, could be evaluated alongside graph-based approaches to potentially enhance performance. An ablation study comparing the contributions of each pipeline component would provide valuable insights into optimizing and refining POMMIX’s architecture.

**Questions:**

How does POMMIX handle biases or gaps in training data, particularly for underrepresented molecular categories? Would augmenting the data or introducing data-driven regularization improve generalization?

What steps could be taken to reduce overfitting, particularly for smaller or highly correlated datasets? Would techniques such as dropout, regularization, or data augmentation improve model robustness?

Have other embedding strategies, such as chemical language models, been considered as alternatives to graph-based methods? Would a hybrid approach provide any advantages, and how might the performance of different embeddings compare within the POMMIX framework?

Has an ablation study been conducted to evaluate the individual contributions of POMMIX’s components (graph neural networks, attention mechanisms, cosine prediction heads)? How would understanding the effectiveness of each component inform future improvements?

---

> ### Author Response · Authors · 2024-11-21
> **We address all reviewer points about dataset size, generalization, and overfitting. We have carefully considered limitations involved in our model and our datasets.**
>
> Thank you for your suggestions and comments about our work. We believe we have significantly improved the submission based on your suggestions, by further justifying that our model architecture truly achieves SOTA performance via thorough ablation studies of the POMMix pipeline. We address your concerns point-by-point below; we hope that our responses will allay your concerns, and that you will consider increasing your score of our submission.
>
> ---
>
> > How does POMMIX handle biases or gaps in training data, particularly for underrepresented molecular categories?
>
> We agree with your observation, and we state the same in our manuscript. We have already presented multiple studies looking at generalization and dataset bias (**Figure 5 and 6**). We believe that our curated datasets are already as comprehensive as possible; further olfaction experimentation is outside the scope of this work and of the conference.
>
> The performance of POMMix, like many data-driven models, is dependent on the quality and diversity of the training data. The limited availability of training data is a recognized challenge in this domain, as we have noted in lines `53 to 54`. This motivates the need for building inductive biases into POMMix to work in this low-data regime.
>
> Regarding generalization, we studied this in our ablation study, seen in **Figure 5**. While we find that mixture sizes do not affect POMMix performance, more data with more diverse molecules (as seen in leave-molecules-out splits) can greatly improve future model performance. Regarding dataset bias, we study possible human biases captured by our model in **Figure 6b**. We have taken care to study the limitations of our model and the dataset.
>
> > Additionally, POMMIX’s hierarchical architecture is computationally intensive, posing scalability challenges for very large datasets.
>
> In general, GNNs and graph transformers are quite scalable, and this has been demonstrated in multiple works: [Graphormer](https://arxiv.org/abs/2106.05234), [MPNN](https://arxiv.org/pdf/1904.01561v5). GNNs allow efficient message-passing between atomic and edge features of the molecules, and have achieved SOTA results on many larger datasets such as [ZINC](https://doi.org/10.1021/acscentsci.7b00572) (250k), and [OGB-LSC](https://arxiv.org/abs/2103.09430) (3.8M). Work with almost 10,000 mixtures from [Zhang et al. (2023)](https://arxiv.org/pdf/2312.16473), more than 10x the data available to us, show that computational cost and scalability issue is not a major concern, given the limited amounts of data (addressed above) and the lightweight nature of POMMix.
>
> > What steps could be taken to reduce overfitting, particularly for smaller or highly correlated datasets? Would techniques such as dropout, regularization, or data augmentation improve model robustness? Would augmenting the data or introducing data-driven regularization improve generalization?
>
> We have indeed implemented regularization strategies, such as early stopping, lower learning rate for pre-trained weights, and dropout layers (stated in **Appendix A.3**). Performance results are averaged over cross-validation sets to prevent overestimating model performance. We perform data ablation to further study generalization abilities (**Figure 5**). Finally, enforcing inductive biases by incorporating knowledge about the problem space into the model architecture ensures that the learned mixture representations closely follow the chemistry of olfaction, and prevent overfitting on our datasets.
>
> We have considered pre-training CheMix with augmented data (see **Appendix A.8**) in which we defined that the perceptual distance between two molecules in the GS-LF dataset would correspond to the Jaccard distance between their odor labels, generating a total of 15571 augmented data points. Our augmentation technique however did not lead to significant changes to the embedding space of the POMMix model (**Figure A2**), and further led to poorer performance of the model (**Table A5**).
>
> We recognize that data augmentation and pre-training in olfactory modeling is an open problem that warrants further investigation. Due to the lack of large amounts of publicly available training data, as stated in our response to your first question, data augmentation would be an ideal strategy to improve the performance of olfactory models. However, as there is currently no strong physical prior for data augmentation, we believe that deploying augmented data in training can lead to poorer performance on unseen mixtures.

---

> ### Author Response · Authors · 2024-11-21
> **We perform additional ablation studies of all aspects recommended by the reviewer, including GNN, chemical language embeddings, attention mechanism, and the prediction head.**
>
> > Have other embedding strategies, such as chemical language models, been considered as alternatives to graph-based methods? Would a hybrid approach provide any advantages, and how might the performance of different embeddings compare within the POMMIX framework?
>
> In order to address the use of chemical language model embeddings, we perform additional experiments with MolT5, a language model trained specifically for molecules and natural language chemical annotations. Results are shown in **Table A4**. The MolT5 embeddings were used with our baseline model XGBoost ($\rho = 0.432 \pm 0.020$), and also with CheMix ($\rho = 0.672 \pm 0.021$). The POM embeddings clearly improve the performance of our model: for example, XGBoost + POM ($\rho = 0.497 \pm 0.041$) and CheMix + POM ($\rho = 0.749 \pm 0.030$). We also refer you to [our response](https://openreview.net/forum?id=6wXYXYSFPK&noteId=1MyLT08qj2) to reviewer `uSgz`, where we evaluated other molecular representation methods (see **Table A2**), and highlighted that GNNs deployed in the POM achieve SOTA performance on olfactory prediction tasks, making it a suitable choice for constructing mixture representations.
>
> > Has an ablation study been conducted to evaluate the individual contributions of POMMIX’s components (graph neural networks, attention mechanisms, cosine prediction heads)? How would understanding the effectiveness of each component inform future improvements?
>
> Thank you for this suggestion. We have performed individual ablation on the POMMix model already in our paper, elaborated in **Table 1**, and **Figure 4**. We show the increase in performance as we introduce the different components. By comparing CheMix with the baselines of XGBoost and the Snitz similarity, we show the effectiveness of the CheMix attention mechanism for mixture modeling.
>
> To further support our claims, we include the results for three additional ablated models. As suggested in your prior comment, we study the use of chemical language model embeddings. We try MolT5 with both XGBoost and CheMix models, and find lower performance than when we use the POM embeddings with the respective models. We further combine CheMix with RDKit features, which again shows lower performance than CheMix with the POM embeddings. This demonstrates the importance of the pre-trained mono-molecular GNN POM in the POMMix mixture embeddings. These additional experiments are shown in **Table A4**.
>
> We further perform ablation studies on the prediction head. We train four additional models of CheMix with different prediction heads: mean + linear, concatenate + linear, PNA-like + linear, and unscaled cosine distance (shown in **Table A3**). We show that using any aggregation method of the mixture embeddings followed by a linear layer performs worse than the unscaled cosine distance when looking at the test correlation coefficients, while these aggregated mixture embeddings achieve RMSE values lower than the unscaled cosine distance prediction head. The scaled cosine distance prediction head combines the strengths of both, achieving the best test performance on all three metrics (Pearson, Kendall, and RMSE). We add the above discussion to **Appendix A.7**.

---

> ### Author Response · Authors · 2024-11-26
>
> Thank you for your feedback on our submission. We have provided further analysis and implemented your suggestions regarding different models and embeddings. We have revised our manuscript accordingly, and provided detailed responses to your concerns about generalizability and overfitting.
>
> We kindly ask you to review the updates and let us know if they resolve your concerns, and please feel free to share any further suggestions.

---

> ### Author Response · Authors · 2024-12-02
>
> We hope you had time to review our responses and look at our corresponding changes. As the discussion period comes to an end, we hope that you can give us any remaining feedback.
>
> If there are no further comments or concerns, we would highly appreciate an increase in the score.

---

### Official Review · Reviewer_uSgz · 2024-11-02

**Soundness:** 3
**Presentation:** 3
**Contribution:** 3
**Rating:** 6
**Confidence:** 2

**Summary:**

In the paper, the authors introduce POMMIX, which is an extending framework of the POM to represent mixtures. The POMMIX framework includes the POM model pretrained with mono-molecular data, CHEMIX attention model and  a cosine distance prediction head. Experiments on the mixture dataset show the empirical performance of the method.

**Strengths:**

1. The presentation of the method is very clear and easy to understand.
2. Many details of the experiments are provided, including the dataset and the schematic of the POMMIX model.

**Weaknesses:**

1. I think ablation study about different POM network architecture is necessary.  Since the GraphNets architecture is not the current SOTA architecture, I think using other architecture may improve the performance, e.g. [1].

[1].Ying, Chengxuan, et al. "Do transformers really perform badly for graph representation?." Advances in neural information processing systems 34 (2021): 28877-28888.

**Questions:**

See the weaknesses part above.

---

> ### Author Response · Authors · 2024-11-21
> **We perform additional studies with SOTA architectures, ablating our graph model. POM architecture is still optimal for our modeling problem.**
>
> We thank the reivewer for their suggestions. We have implemented other graph-based models in order to understand the significance of using our current GNN for the POM embedding and the POMMix model. This strengthens our claims and our work using POMMix for olfactory mixture modelling. We hope that the additional experiments we performed in response to your comment are satisfactory, and that you will consider increasing your score of our submission.
>
> ---
>
> > I think ablation study about different POM network architecture is necessary. Since the GraphNets architecture is not the current SOTA architecture, I think using other architecture may improve the performance, e.g. [1].
> >> [1].Ying, Chengxuan, et al. "Do transformers really perform badly for graph representation?." Advances in neural information processing systems 34 (2021): 28877-28888.
>
> We agree that investigating other state-of-the-art (SOTA) graph architectures is valuable. We conducted experiments with graph transformer models on the GS-LF dataset: Graphormer (as suggested), and the GPS model (which currently achieves SOTA on the ZINC and OGB datasets). We provide a comparison with our GraphNets-based POM.
>
> We experimented with the Graphormer (slim) and GPS models, achieving validation AUROC $0.856$ and AUROC $0.864$, respectively. While we acknowledge that the aforementioned graph transformers can achieve SOTA performance on certain molecular modeling tasks, our results indicate that our GraphNets-based POM performs competitively, and even outperforms (AUROC $0.884$), the more complex architectures in our specific applications, and in such low data (< 5000 molecules) regimes.
>
> Furthermore, [recent work](https://arxiv.org/pdf/2406.08993) supports the competitive nature of classic GNNs with graph transformers. For such small data regimes, the increased expressivity of the graph transformer architecture may result in overfitting and decreased test performances. More recent work from [Shin et al. (2024)](https://www.researchsquare.com/article/rs-3607229/v1) modeling the GS-LF dataset with transformer-based models show that the POM from [Lee et al. (2023)](https://doi.org/10.1126/science.ade4401) is still SOTA  -- we note that their results with GNN-based features are incongruent with previously reported metrics and cannot be reproduced due to a lack of code availability. We have added the results of the additional graph-based models in **Table A2** along with the above discussion in **Appendix A.7**.

---

> > ### Comment · Reviewer_uSgz · 2024-11-25
> >
> > Thanks for your reply and I am satisfied with the responses. I will keep my positive score.

---

### Official Review · Reviewer_XFHX · 2024-11-03

**Soundness:** 3
**Presentation:** 4
**Contribution:** 3
**Rating:** 6
**Confidence:** 4

**Summary:**

The authors report POMMIX, an approach to extend principal odor maps to mixtures. For this, they derive embeddings using a GNN and then use attention to aggregate those embeddings. Since they focus on a contrastive task, they use cosine predictive heads.
They show that their approach outperforms various baselines.

**Strengths:**

- The paper is very well written and the methodology is clearly described
- The authors also carefully described their hyperparameter optimization
- It also seems as if the authors were careful in building baselines
- Dealing with mixtures is an important problem in chemistry that is often ignored - many people focus on predicting the properties of pure compounds as this is simpler
- The analysis of the "White noise hypothesis" is a nice case study

**Weaknesses:**

- The methodology the authors proposed seems to be well-suited to address the task, but there seem to be no major innovations. Using GNN-derived embeddings and aggregating them via attention has been done before, e.g., in https://arxiv.org/pdf/2312.16473
- The attention maps are interesting, but I found it difficult to gain insights from them. Also the discussion in the paper is mainly focussed on general observations as number of interactions increasing with the number of components.

Overall, it is an interesting applied ML paper that nicely shows how ML can be applied to an exciting chemistry problem. There is little advancement in the ML methodology.

**Questions:**

- Perhaps it goes beyond the scope of the work but I wonder if the performance of such models might not be improved a lot with MixUp-like augmentation techniques.
- Could one obtain more chemical insights from the attention-map analysis by analyzing, for instance, how often certain functional groups/scaffolds interact with each other?

---

> ### Author Response · Authors · 2024-11-21
> **We clarify our contributions relative to related works on modeling molecular mixtures.**
>
> We thank the reviewer for their thoughtful comments and suggestions, which have improved our manuscript. We provide additional studies and analysis into the interpretability of our model. We also further define our contributions and the uniqueness of the problem we are modeling: compiling olfactory mixture dataset, designing inductive biases into the model, multi-step pre-training and fine-tuning, and interpreting the mixture representation through analysis of attention and physical phenomena. We address your concerns point by point below.
>
> ---
>
> > The methodology the authors proposed seems to be well-suited to address the task, but there seem to be no major innovations. Using GNN-derived embeddings and aggregating them via attention has been done before, e.g., in https://arxiv.org/pdf/2312.16473
>
> We appreciate your feedback and would like to clarify the differences between MolSets and our work, which goes beyond simply using GNNs and attention. While both works utilize these components, our approach is tailored to the complexities of olfactory mixture representation, and presents novel advancements in the following ways:
> - Our work tackles a much lower data regime (~760 mixtures vs. ~10000 in MolSets), which necessitates the pre-training of POM with mono-molecular olfaction data (also limited, ~5000 molecules), and pre-training of the CheMix, followed by end-to-end training of the full POMMix model. Our multi-step training introduces inductive biases and regularizes each individual network.
> - Our work also tackles a dataset with larger and more complex mixtures: up to 43 compounds, variable size vs. 4 compounds, fixed size, with many molecules that are “unimportant” to the human olfaction.
> - Our modeling exploration for mixture modeling is more extensive: different pre-training strategies, different aggregation methods with different symmetries, different styles of attention (softmax, sigmoid), and more baselines (including new baselines with SOTA graph models in **Table A2**).
> - We conduct comprehensive ablation studies to demonstrate the contribution of each inductive bias and design choice. We separate the contributions of the POM embeddings, from the attention mechanism. We’ve also added additional experiments to study the effects of the prediction head (in **Table A3**).
> - Our work goes beyond property prediction; our dataset and POMMix allows us to learn distance-aware mixture representations. Our model has an additional hierarchy of comparison between the mixture embeddings, which has its own symmetries associated with it. This is an important step in the digitization of olfactory space, which has not  been previously done before.
> - We study interpretability for mixtures, specifically utilizing a sigmoid attention mechanism. This offers insights into the drivers of mixture perception, and the relation to mono-molecular contributions, a feature not explored in MolSets. We have added additional analysis (as requested by your next comment) to our revised submission (**Appendix A.10**).
> - We further use the learned representations to explore physical olfactory phenomena. These studies are unique to our work and provide a deeper understanding of olfaction itself through our model POMMix, differentiating it from MolSets’ focus on property prediction. We investigate:
>   - The white noise hypothesis.
>   - Generalization to unseen molecules and different mixture sizes.
>   - Human biases in perception data, and the effects on POMMix.
>
> We have added the work of MolSets to the *Related works* section.

---

> ### Author Response · Authors · 2024-11-21
> **We perform additional analysis on the interpretation of attention weights of our model, and how it translates to individual molecular smells and chemical structures. We also address augmentation methods.**
>
> > The attention maps are interesting, [...].
> > Could one obtain more chemical insights from the attention-map analysis by analyzing, for instance, how often certain functional groups/scaffolds interact with each other?
>
> Thanks for your suggestion. The physical interpretation of transformer-based architectures applied to chemical problems is a longstanding challenge, and we would like to emphasize that our work on mixture self-attention is preliminary. Fully addressing this question would be another work in itself. Nevertheless, we agree our analysis would benefit from further interaction investigation at the chemical structure-level.
>
> To complement our original analysis, we performed additional analysis to provide a general view of what “interacting” and “non-interacting” molecules look like by deriving heuristics across the entire set of unique mixtures. Within each mixture, we looked at the key-ed molecules associated with the minimum and maximum attention weight for each query molecule, focusing on queries interacting “strongly” with a key. We visualized with UMAP the POM embeddings projected by CheMix through one linear layer of key molecules exclusively found as maximizing/minimizing attention weights with GS-LF labels and observed a clear separation in the embedding space between the two classes (see **Appendix A.10, Figure A5**). This suggests certain types of molecules are prioritized/deprioritized when it comes to updating the molecular embeddings within a mixture. We then conducted hierarchical clustering of these molecules based on the Jaccard similarity of their GS-LF labels. We note that molecules within clusters are generally either “strongly” or “weakly” interacting. We selected a few representative molecules for each of the clusters and noticed strong structural differences between them. We observe that ester/aldehydes with long alkane chains tend to be labeled as “non-interacting” keys, while sulfur-containing molecules and molecules containing aromatic rings tend to be labeled as “interacting” ones. This corroborates what we see in the attention heatmaps (**Figures 7, A3, A4**), in which molecules with distinct olfactory characteristics (i.e., garlicky, sulfurous etc.) receive more attention and have strong interactions with query molecules when used to distinguish chemical mixtures.
>
> > Perhaps it goes beyond the scope of the work but I wonder if the performance of such models might not be improved a lot with MixUp-like augmentation techniques.
>
> Thanks for the suggestion. While MixUp augmentations were designed for classification, we believe the idea of interpolating between mixture vectors to generate more training data merits further investigation and discussion. Since MixUp-like augmentation is meant to enforce the inductive bias that interpolating between feature vectors leads to linear interpolations of the associated targets, we believe that our model architecture (via CheMix) already incorporates this inductive bias. Specifically, since the perceptual similarities are computed from the cosine distance between high-dimensional mixture representations, and the mixture embedding space should already be organized in such a way that interpolations between mixture embeddings directly return their corresponding difference in the cosine distance space. We further show below, as a response to another reviewer for a requested ablation study, that the cosine prediction head is necessary for the model’s good performance. We had previously come across a [MixUp-like data augmentation strategy](https://www.synapse.org/Synapse:syn61941777/wiki/629245) in olfactory modeling, in which the following was considered:
>
> - If $M_1$ has molecule $A$ but $M_2$ does not, adding $A$ to $M_2$ increases their explicit similarity by $k_1$.
> - If $M_1$ has molecule $A$ and $B$, but $M_2$ does not, adding both $A$ and $B$ to $M_2$ increases explicit similarity by $2k_1$
> - If $M_1$ and $M_1$ have molecule $C$, removing $C$ decreases their explicit similarity by $k_2$.
>
> The constants $k_1$ and $k_2$ were determined in this study via hyperparameter tuning as there were no physical priors to the impact of how these molecular additions or removals would impact the perceptual similarity. However, we believe that augmenting the data in this way could cause extreme overfitting, especially since the mixture dataset is less than 1000 points.
>
> We also refer you to [our response](https://openreview.net/forum?id=6wXYXYSFPK&noteId=IkTmn7nTaR) to reviewer `BX4L`, where we discuss data augmentation generally in further detail.

---

> ### Author Response · Authors · 2024-11-26
>
> Thank you for your feedback on our submission. We have revised the manuscript and provided a detailed response to address your concerns.
>
> We kindly ask you to review the updates and let us know if they resolve your comments. Your insights have been invaluable, and we truly appreciate your time and effort. Please feel free to share any further suggestions.

---

> > ### Comment · Reviewer_XFHX · 2024-11-28
> >
> > The additional experiments added useful context, and I added my score accordingly.
> >
> > For an even higher score, I think a more direct comparison to MolSets (and comparable approaches) would be good (i.e., to quantify that there is a positive impact from your statement, "Our modeling exploration for mixture modeling is more extensive").

---

> ### Author Response · Authors · 2024-11-29
> **Direct comparison with MolSets, and additional experiments with MolSets.**
>
> Thank you for your comments and for increasing your score. We agree that a more direct comparison with MolSets would be valuable.
>
> We have trained MolSets with our olfaction data to provide a direct and quantifiable comparison with our model. We note that because MolSets is purpose-built for their problem (regression for predicting conductivity of a mixture), re-optimizing their proposed architecture for our problem is non-trivial. Therefore, we had to make various compromises to implement their architecture to our case:
> - Our data does not contain weight fractions and dilutions, and these were set to 1.0.
> - In our data filtering pipeline, we intentionally removed salts and multimolecular SMILES, and thus we had to remove their model component that accounts for the salts.
> - As MolSets directly predicts with one mixture embedding, we had to generate two mixtures to get two MolSets embeddings, which are then concatenated and run through their MLP predictor head.
>
> On our cross-validation test splits, MolSets (using SAGEConv) achieves $\rho = 0.418 ± 0.063$, and MolSets (using DMPNN) achieves $\rho = 0.329 ± 0.092$. MolSets with SAGEConv was the best model achieved by Zhang et al., however, our POMMix architecture still achieves better test results ($\rho = 0.779 ± 0.028$). We finally note that these necessary compromises may have led to poorer performance of MolSets on our problem.
>
> Additionally, we hope that this qualitative direct comparison of the modeling efforts between our work and MolSets appropriately highlights the extensive efforts we have undertaken to explore the impacts of various model design choices for olfactory mixture modeling.
>
> As the upload period is over, we will add these analyses to a later version of our manuscript.
>
>
> | Feature                                | POMMix                                                                         | MolSets                                                              |
> |----------------------------------------|--------------------------------------------------------------------------------|----------------------------------------------------------------------|
> | Task                                   | Similarity between embeddings                                                  | Regression from embeddings                                           |
> | Molecular Representation               | Graph neural network, RDKit molecular descriptors, MolT5 embeddings            | Graph neural network                                                 |
> | Pre-training                           | Yes, on odor descriptor prediction. Yes, on the GNN embeddings for CheMix model | None                                                                 |
> | Molecular Attention                    | Self-attention                                                                 | None                                                                 |
> | Convolution Operators Tested           | GraphNets (GATConv + FiLM + PNA), Graphormer, GPS, DMPNN                       | SAGEConv, GraphConv, GCNConv, GATConv, DMPNN                         |
> | Mixture Representation                 | Permutation-invariant aggregation of molecular representation                  | Permutation-invariant aggregation of molecular representation        |
> | Molecular Representation Aggregations  | Attention, PNA-like, mean, cross-attention                                     | Attention + weighted-sum, weighted-sum, concatenation                |
> | Mixture attention Methods evaluated    | Softmax, Sigmoid                                                               | Softmax                                                              |
> | Proportion Representation              | None (not available in dataset)                                                | Weighted-sum proportional to weight-fraction of component in mixture |
> | Prediction Head                        | Scaled cosine, unscaled cosine, mean + MLP, concatenate + MLP, PNA-like + MLP  | MLP                                                                  |
> | Batching                               | Yes                                                                            | No, architecture only permits SGD                                    |

---

### Author Response · Authors · 2024-11-21

Dear Reviewers,

Thank you for your valuable feedback. We have carefully addressed your concerns and implemented your suggestions where possible and produced additional results to strengthen our claims.

We paraphrase the comments we have included in our updated PDF in the official comments to each reviewer. We will summarize the reviewer comments, the additional experiments, and the corresponding changes below.

- **Ablation of features, GNN, CheMix, and prediction head**. We provided ablation studies (in addition to **Table 1**) on the different parts of the POMMix model, as suggested by Reviewer `uSgz`  and `BX4L`. All experimental results are found in the **Appendix A.7** section.
  - GNN ablation, comparing our models with other SOTA graph models. (**Table A2**)
  - CheMix prediction head ablation, justifying our choice of the scaled cosine distance prediction head (**Table A3**)
  - Ablation of feature, using chemical language model embedding and compare results between baseline and CheMix. (**Table A4**)

- **Possible augmentation methods for our dataset**. As brought up by Reviewers `XFHX` and `BX4L`, augmentation methods may improve the performance and generalization of our model.
  - We have previously done augmentation of mixture data with single-molecule mixtures attained from the larger GS-LF, augmenting our dataset by ~15k mixtures.
  - We add **Table A5** to show the reduced performance of our CheMix attention model from this augmentation.
  - MixUp augmentation is not applicable to our modeling problem. Other augmentation methods require hyperparameter tuning on the existing dataset in order to generate new similarity; however there is no physical prior to motivate doing this, and would likely cause overfitting.

- **Further analysis on the chemistry of interpretability study**. As brought up by Reviewer `XFHX`, more insight into the interpretability study would make the work stronger, and also more distinct from other work in mixture modeling.
  - We provide chemical structure insights based on the interpretability studies (**Appendix A.10**) by clustering the molecules based on odors, and looking at the correspondance of the clusters with the strength of interaction.
  - Additional plots are added in **Figure A5**.

We appreciate your time and effort in providing constructive feedback for our submission.

Kind regards, the authors.

---

### Meta-Review · Area_Chair_rEXz · 2024-12-26

**Metareview:**

The paper extends POM (primary odor map) of single compounds molecules to POMMIX handling mixture compounds mixtures olfactory properties.  The molecular embedding are obtained via a graph neural networks , the mixture representation is obtained via an attention aggregation, and  a cosine prediction head encodes perceptual distance in the mixture embedding space. State of the art results are obtained on various benchmarks.

Some of the reviewers did not unfortunately engage during the discussion phase although reminded several times by the AC and they did not respond in the AC/reviewers discussion phase.
Given this lack of interaction, I have checked the paper and the reviews and author rebuttal and building my decision on that.
In the rebuttal phase:
* Authors provided in the rebuttal comparison to Molsets
* Authors provided abalations on GNN, chemical language embeddings, attention mechanism, and the prediction head.

One of the main concerns in the paper is that the metholdolgy is not indeed novel as raised by the reviewer XFHX, hence the ablation of the method and more details on it will further strengthen the work.
Unfortunately, although they were provided in the discussion phase, these were not incorporated to the manuscript and it is hard to judge the paper in its current form and a major revision is needed.

**Additional Comments On Reviewer Discussion:**

Please see above for how the rebuttal and  discussion phase of the paper.

---

### Decision · Program_Chairs · 2025-01-22

Reject